# The Respiratory Burst Oxidase Homolog Protein D (*GhRbohD*) Positively Regulates the Cotton Resistance to *Verticillium dahliae*

**DOI:** 10.3390/ijms222313041

**Published:** 2021-12-02

**Authors:** Wanting Huang, Yalin Zhang, Jinglong Zhou, Feng Wei, Zili Feng, Lihong Zhao, Yongqiang Shi, Hongjie Feng, Heqin Zhu

**Affiliations:** 1State Key Laboratory of Cotton Biology, Institute of Cotton Research, Chinese Academy of Agricultural Sciences, Anyang 455000, China; hwanting2020@163.com (W.H.); zhangyalin@caas.cn (Y.Z.); zhoujl510@163.com (J.Z.); weifeng@caas.cn (F.W.); fengzili@caas.cn (Z.F.); zhaolihong@caas.cn (L.Z.); 13619832007@163.com (Y.S.); 2Zhengzhou Research Base, State Key Laboratory of Cotton Biology, School of Life Science, Zhengzhou University, Zhengzhou 450001, China; 3School of Agricultural Sciences, Zhengzhou University, Zhengzhou 450001, China

**Keywords:** cotton, Verticillium wilt, *V. dahliae*, resistance gene, ROS, GhRbohD

## Abstract

Verticillium wilt, mainly caused by a soil-inhabiting fungus *Verticillium dahliae*, can seriously reduce the yield and quality of cotton. The complex mechanism underlying cotton resistance to Verticillium wilt remains largely unknown. In plants, reactive oxygen species (ROS) mediated by Rbohs is one of the earliest responses of plants to biotic and abiotic stresses. In our previous study, we performed a time-course phospho-proteomic analysis of roots of resistant and susceptible cotton varieties in response to *V. dahliae*, and found early differentially expressed protein burst oxidase homolog protein D (*GhRbohD*). However, the role of *GhRbohD*-mediated ROS in cotton defense against *V. dahliae* needs further investigation. In this study, we analyzed the function of *GhRbohD*-mediated resistance of cotton against *V. dahliae* in vitro and in vivo. Bioinformatics analysis showed that GhRbohD possessed the conservative structural attributes of Rbohs family, 12 members of RbohD out of 57 Rbohs in cotton. The expression of *GhRbohD* was significantly upregulated after *V. dahliae* inoculation, peaking at 6 hpi, and the phosphorylation level was also increased. A VIGS test demonstrated that ROS production, NO, H_2_O_2_ and Ca^2+^ contents of *GhRbohD*-silenced cotton plants were significantly reduced, and lignin synthesis and callose accumulation were damaged, important reasons for the impairment of *GhRbohD*-silenced cotton’s defense against *V. dahliae*. The expression levels of resistance-related genes were downregulated in *GhRbohD*-silenced cotton by qRT-PCR, mainly involving the lignin metabolism pathway and the jasmonic acid signaling pathway. However, overexpression of *GhRbohD* enhanced resistance of transgenic *Arabidopsis* to *V. dahliae* challenge. Furthermore, Y2H assays were applied to find that GhPBL9 and GhRPL12C may interact with GhRbohD. These results strongly support that *GhRbohD* activates ROS production to positively regulate the resistance of plants against *V. dahliae*.

## 1. Introduction

Cotton, a primary natural fiber crop, is of great importance to the global textile industry [1,2]. Cotton Verticillium wilt is one of the main dangerous diseases globally, which poses a serious threat to the sustainable development of cotton production [3,4,5,6]. Verticillium wilt of cotton is a kind of soil-borne vascular bundle disease caused by *Verticillium dahliae*, which can cause vessel blockage, leaf yellowing and wilting, and even plant death, mainly resulting in yield and fiber quality loss in cotton [4,7]. Fungus invades from the roots and systematically infects plants, and the number of dormant structure microsclerotia directly determines the incidence of the host which is very difficult to control [6,8]. In addition, it is very difficult to obtain Verticillium wilt disease-resistant cotton varieties by traditional breeding methods. Thus, it is essential to identify cotton disease resistance genes, especially in the early stage of disease resistance, and incorporate them into elite cotton cultivars. With the development of genetics, molecular biology and genetic engineering, these methods will accelerate progress in the resistance of plants to Verticillium wilt [1]. Ultimately, further understanding is still needed of the key genes in the resistance mechanism and regulation mechanism of cotton against Verticillium wilt [6].

The accumulation of ROS is generally found in systemic disease resistance in many plants [9]. ROS acts as signal molecules, transmitting signals after pathogen invasion, causing plants to produce a series of defense responses such as Ca^2+^ signaling, kinase cascades, hormone signaling and so on [10]. ROS also has a negative effect in plants, and excessive accumulation of ROS will cause harm, so plants have to tightly regulate the stability of ROS [11]. Plants need to deal with ROS produced by aerobic respiration, including superoxide anion radicals (O^2−^), hydrogen peroxide (H_2_O_2_) and hydroxyl radicals (-OH) [12]. Moreover, ROS outbreak, as a conservative immune response, is mainly mediated by NADPH oxidase (NOXs), which is also called Rbohs in plants [13]. NADPH oxidase enzyme generates ROS, with a six-transmembrane-domain structure and C-terminal region, a conservative NADPH- and FAD-containing motif, and a N-terminal region with a two EF-hand motif and phosphorylation site [14]. Mounting evidence indicates that organisms have established their own regulatory system in the process of evolution, and the production of ROS mediated by Rbohs plays a crucial role in plant development, as well as biological and abiotic stress responses [15].

Previous studies have shown that there were ten kinds of Rboh in *Arabidopsis thaliana*, from AtRbohA to AtRbohJ, among which AtRbohD was the most characteristic, described as a key component of plant defense in *Arabidopsis* [13,16]. AtRbohD produces superoxide anion radicals in *A. thaliana* in response to intracellular calcium and phosphorylation signals, activating the generation of ROS in the response of *Arabidopsis* to all pathogens tested [15,17,18,19]. Previous research has shown that phosphorylation of RbohD at the C or N terminal was necessary for plant resistance to pathogens and conserved across plant lineages [20,21,22]. Moreover, the N-terminal region of RbohD was phosphorylated by a variety of protein kinases, which were synergistically activated by Ca^2+^ binding and protein phosphorylation to produce ROS, playing various roles in plants and participating in biological and non-biological stress responses [23].

Here, we identified and investigated the role of *GhRbohD*-mediated ROS in cotton defense against *V. dahliae*. Silencing the *GhRbohD* gene compromised cotton resistance to fungus and weakened ROS production, and NO, H_2_O_2_ and Ca^2+^ contents, with lignin synthesis and callose accumulation also impaired. Furthermore, the expression levels of resistance-related genes mainly involving the lignin metabolism and jasmonic acid signaling pathways were significantly reduced in *GhRbohD*-silenced cotton. In contrast, overexpression of *GhRbohD* improved resistance to *V. dahliae* in *Arabidopsis* plants. Collectively, our study provides important evidence illustrating that *GhRbohD* positively regulates plant resistance to Verticillium wilt.

## 2. Results

### 2.1. Gene Clone and Bioinformatic Analysis of GhRbohD

In our previous study, a time-course phospho-proteomic analysis of roots between resistant and susceptible cotton lines infected with *V. dahliae* was performed [24]. A respiratory burst oxidase homologue *GhRbohD* (Ghir_A05G026340) was identified from the plant–pathogen interaction pathway, which may be related to plant resistance to Verticillium wilt. We cloned the full-length cDNA of *GhRbohD* from cotton cultivar Zhongzhimian 2, which consists of 2793 bp, and encodes a protein of 930 aa with six trans-membranes, an N-terminal containing the EF-hands and a conserved C-terminal region containing the NADPH and FAD binding motifs. In addition, four obviously phosphorylated sites were found at the N-terminal of *GhRbohD* after inoculation with *V. dahliae*, located in sites 11, 19, and 22 of serine and 17 of threonine.

To investigate their evolutionary relationships, the phylogenetic analysis of the 57 Rbohs from the three cotton species (*G. hirsutum*, *G. raimondii* and *G. arboreum*) and 10 Rbohs from *Arabidopsis thaliana* were performed by ClustalX2.1 and MEGAX5.0 software. According to homology, the 57 Rbohs in cotton were divided into 7 branches (named RbohA, B, C, D, E, F, H) (Figure 1). There were 12 members in each of RbohA and D, 4 members in each of RbohB and H, 13 members in RbohC, 7 members in RbohE and 5 members in RbohF. *G. hirsutum* (AD1, upland cotton, HAU assembly) included six kinds of *GhRbohD*.

The conserved motif analysis of the Rbohs family in *G. hirsutum* showed that the Rbohs homologous genes in *G. hirsutum* were highly similar and contained at least 10 conserved motifs named motif 1–10 (Appendix A). To analyze the exon-intron distributions of the Rbohs genes, the gene structure display server was utilized, and these results indicated that the Rbohs in *G. hirsutum* had similar gene structure and conserved motifs. Chromosomes localization analysis of Rbohs on *G. hirsutum* showed that all 30 Rboh genes were distributed in 18 chromosomes. *GhRbohD* (Ghir_A05G26340.1) was mapped on chromosome A05 (Appendix A). The promoter sequence of Rboh family genes were extracted and cis-acting elements were analyzed to obtain cis-acting elements related to disease resistance, associated with defense and stress, Abscisic acid, Salicylic acid, MeJA, and Auxin (Appendix A).

### 2.2. Expression Pattern of GhRbohD Gene

The expression pattern of the *GhRbohD* gene showed that the expression of *GhRbohD* was generally of high volume in cotton plants (Figure 2A). Among them, the anther, pistil, root, sepal and torus were all about 10 to 20 times, and the bract, petal, and stem 20–30 times. By observing the changes in transcription levels of Rbohs family genes in *G. hirsutum* inoculated with high pathogenicity (VD991) and low pathogenicity (VD07038) *V. dahliae*, the expression level of *GhRbohD* increased by about five times after infection by *V. dahliae* (Figure 2B). The roots of cotton infected with *V. dahliae* strain Vd080 at 0, 1, 6, 12 and 24 h post inoculation (hpi) were stored, respectively. The expression level of *GhRbohD* gene in cotton was determined by qRT-PCR. The results showed that the expression level of *GhRbohD* gene was upregulated rapidly and increased significantly after pathogen infection, especially at 6 hpi, indicating that this gene may play an important role in the response of cotton to the stress of *V. dahliae* (Figure 2C). Furthermore, we performed the analysis of the levels of protein phosphorylation modification in recombinant inbred lines’ (RIL) susceptible/resistant varieties under pathogenic stress [24]. In susceptible lines (RIL-S), the phosphorylation level after inoculation was 1.5 times compared to without inoculation, and was nearly 2 times in resistant lines (RIL-R) at 1 dpi (Figure 2D), which showed that the phosphorylation level of GhRbohD significantly increased both in susceptible and resistant varieties after infection with *V. dahliae*.

### 2.3. Silencing of GhRbohD Reduced the Basal Resistance of Cotton against V. dahliae

In order to explore the role of *GhRbohD* in defense against *V. dahliae* infection, the leaves of *GhRbohD*-silenced cotton plants with a positive control gene (*GhPDS*) showed significant whitening, which indicated that the TRV-VIGS system worked well in cotton plants (Figure 3A). The expression of *GhRbohD* in the silenced cotton plants was significantly lower compared to the control plants (Figure 3B). Plant disease resistance assessment demonstrated that the yellowing, collapse and even the fall off of leaves in *GhRbohD*-silenced plants were more serious than those in the control plants, and the DI value of *GhRbohD*-silenced plants was 59.06 ± 3.12 which significantly increased by 53.16% compared to the control plants (38.56 ± 3.61) (Figure 3C,D). Compared with *TRV::00* plants, the browning degree of vascular bundle in *TRV::GhRbohD* plants showed significant aggravation (Figure 3E). Moreover, lignin synthesis and callose accumulation were detected, and the results showed that the accumulation of lignin and callose in *GhRbohD*-silenced plants was significantly lower than those in the control plants after *V. dahliae* inoculation (Figure 3F). To further investigate the effects of *GhRbohD* on plant resistance, pathogen hyphae recovery growth from cutting sections of the infected stem was assayed, and the fungal colonization in *TRV::GhRbohD* plants were compared further with *TRV::00* plants (Figure 3G). These results indicated that inhibition of *GhRbohD* gene expression reduced the resistance of cotton against *V. dahliae*.

Accumulating evidence indicates that a series of systematic resistance, such as accumulation of ROS, NO, etc., play crucial roles in a variety of biotic and abiotic stresses. DAB staining demonstrated that the active oxygen burst of *GhRbohD*-silenced plants leaves obviously decreased compared to that of control plants’ leaves (Figure 4A). The H_2_O_2_ contents of *GhRbohD* gene silenced plants were significantly lesser than that of *TRV::00* control plants, with the maximum decrease observed at 12 hpi (Figure 4B). NOA1 is a gene related to NO synthesis. After pathogen inoculation, the expressions of *GhNOA1* in *TRV::GhRbohD* plants were generally lower than that in *TRV::00* control plants (Figure 4C). Besides, the NO contents in the *GhRbohD*-silenced samples decreased significantly over the entire time period (Figure 4D). In addition, the calcium ion (Ca^2+^) contents in the silencing *GhRbohD* plants’ roots were significantly lower compared to the control plants, indicating that Ca^2+^ concentration was positive correlated with GhRbohD activity (Figure 4E).

### 2.4. Silencing of GhRbohD Attenuated the Expression of Resistance-Related Genes

To further elucidate the effect of *GhRbohD* on Verticillium wilt resistance in cotton plants, we monitored the expressions of six resistance-related genes, the results showing that the expression levels of these disease resistance-related genes were generally impaired in *GhRbohD*-silenced plants (Figure 5). As key genes of the lignin metabolism pathway, the gene expression of basic chitinase (*GhCHI*), phenylalanine ammonia lyase (*GhPAL5*) and cinnamic acid hydroxylase (*GhC4H1*) was significantly downregulated in the silencing *GhRbohD* plants compared to the control. In detail, the expression of *GhCHI* reached its valley at 9 hpi, only about one-fifth of the control, and the same as the expression of *GhPAL5*. However, the expression of *GhPAL5* was continuously suppressed in *GhRbohD*-silenced plants compared with the control throughout the monitoring process. Besides, in *GhRbohD*-silenced plants, the expression of resistance-related gene 3 (*GHPR3*) and hypersensitivity marker gene (*GhHIN1*) were also suppressed to varying degrees, especially after 9 hpi, then maintained stable levels. As an important gene of the jasmonic acid pathway, the expression of *GhJaz1* continued to decrease and reached its lowest value at 48 hpi, accounting for just one-sixth of that of the control.

### 2.5. Overexpression of GhRbohD Enhanced Resistance to V. dahliae in Transgenic Arabidopsis

To further detect whether *GhRbohD* confers resistance to *V. dahliae*, an overexpression strategy in *Arabidopsis* plants was used. We constructed the overexpression vector of *GhRbohD* and obtained the transgenic *Arabidopsis* plants by floral-dip method. Three methods were used to identify homozygous *GhRbohD*-overexpressing transgenic lines, including 0.1% kanamycin-screening, PCR and qPCR detection (Appendix A). At 14 dpi, wild-type (WT) *Arabidopsis* showed more severe yellowing and wilting than the OE-*GhRbohD* plants on MS medium (Figure 6A). Besides, plant roots’ infection results showed that the amount of hyphae attachment around the roots of WT *Arabidopsis* were significantly increasing compared to those of OE-*GhRbohD* plants after *V. dahliae* inoculation (Figure 6B). At 15 dpi, the leaves of *Arabidopsis* began to show wilting and yellowing symptoms, compared to the WT plants, the OE-*GhRbohD* plants showed much weaker symptoms at 21 dpi, and the DI value of OE-*GhRbohD* plants was 23.05 ± 1.51, reduced by 55.09% compared with WT plants (51.32 ± 2.12) (Figure 6C,D). This evidence further supports that *GhRbohD* can positively regulate plant resistance to *V. dahliae*.

### 2.6. Subcellular Localization of GhRbohD

To gain direct evidence for *GhRbohD* subcellular localization, tobacco leaves injected with *Agrobacterium* carrying GFP or *GhRbohD*-GFP vectors were observed under a confocal laser microscope for fluorescence detection. The results demonstrated that GFP was uniformly distributed in all parts of the cells, indicating normal operation in this experiment, while *GhRbohD*-GFP was located in the cell membrane and stomata (Figure 7A). Besides, onion epidermal cells were used to further determine *GhRbohD* localization, which also validated that *GhRbohD*-GFP fluorescence was observed in the cell membrane before or after plasmolysis (Figure 7B,C).

### 2.7. GhPBL9 and GhRPL12C May Interact with GhRbohD in Cotton

In order to explore the proteins that may interact with GhRbohD, two proteins GhPBL9 (Ghir_A10G022420) and GhRPL12C (Ghir_A09G015860) were investigated by Y2H. PBL9 is a serine threonine protein kinase, located in the mitochondria and cytoplasm membrane, and its main functions in *Arabidopsis thaliana* include ATP Bing and serine/threonine protein kinase activity. RPL12C is a 60s ribosomal protein, which is localized in the cytoplasm and ribosomes and is related to the coding translation of ribosomal large subunits. To further demonstrate the interaction between those two proteins and GhRbohD, we verified the interaction proteins. Firstly, the target protein, positive control, negative control and the interacting proteins were co-transformed into the yeast medium DDO (SD/-Leu-Trp) and the colony growth was observed. Then the yeast rotation experiment of the interacting proteins was verified on the yeast medium QDO (SD/-Leu-Trp-His-Ade), and the interaction between GhPBL9 and GhRbohD, GhRPL12C and GhRbohD were all observed in 10^−1^, 10^−2^ and 10^−3^ dilutions of yeast liquid (Figure 8). These results showed that GhPBL9 and GhRPL12C may interact with GhRbohD, and the protein interaction was stronger between GhRPL12C and GhRbohD than that between GhPBL9 and GhRbohD.

## 3. Discussion

Currently, Verticillium wilt of cotton caused by *V. dahliae* remains a heavy hindrance to cotton production, the main reason being lack of understanding of the mechanisms of effectively utilizing molecular and genetic engineering techniques to further develop resistant cultivars [6,7]. With in-depth application of technologies such as genome, transcriptome and protein modification (phosphorylation, ubiquitination, acetylation) to analyze cotton disease-resistance genes function and regulation mechanisms for Verticillium wilt, cotton disease-resistance research has made progress. In our previous study, a time-course phospho-proteomic analysis of roots of resistant and susceptible cotton lines in response to *V. dahliae* was performed, finding 30 early differentially expressed proteins, including GhCDPK28, GhCML41 and GhRbohD [24]. Notably, the phosphorylation levels of GhRbohD were significantly different between the resistant and susceptible varieties after inoculation with *V. dahliae* (Figure 2D). In addition, the expression level of *GhRbohD* gene was significantly increased when plants were infected with pathogen at 6 hpi, with upregulating of 50 times compared to the control, indicating that this gene may play an important role in the response of cotton to the stress of *V. dahliae* (Figure 2C). Numerous evidence has shown that the functional identification of earlier resistance genes was a promising way to prevent Verticillium wilt [8,25]. When plants were attacked by pathogens, this induced pathogen-associated molecular pattern (PAMP)-triggered immunity including ROS production, lignin and callose accumulation, elicitation of PR genes and hormone content changes [26,27]. In this study, the VIGS experiment showed that silencing *GhRbohD* in cotton, ROS production, NO and H_2_O_2_ contents were all significantly reduced (Figure 4). These results indicate that as an early disease-resistance gene, *GhRbohD* is involved in the prophase defense response of cotton against *V. dahliae*.

As a conserved signaling output during immunity across host plants and pathogens, ROS plays an important role in plant growth and development, with direct anti-microbial characteristics, but also serves as signaling molecule to activate the next immune response. Furthermore, different concentrations of ROS have a double-edged sword function in plants [11,28,29]. Although pathogen-induced ROS production has been documented for nearly 40 years [30], the detailed mechanism underpinning this activation remains largely unknown. Generally, ROS production is mediated by NADPH oxidase or rboh proteins, and a great deal of evidence clearly illustrates that RBOHD is mainly controlled by Ca^2+^ via direct binding to EF-hand motifs and phosphorylation by Ca^2+^-dependent protein kinases [16,31,32,33]. However, the functional analysis of RBOHD in cotton is far from adequate. In order to further clarify the relationship between ROS, GhRbohD and Ca^2+^ content, the in vitro experiment confirmed that the control plants had more ROS production than the *GhRbohD*-silenced plants, and Ca^2+^ concentration was also significantly higher than that of silent plants (Figure 4E), these results demonstrating the direct correlation between *GhRbohD* gene expression and Ca^2+^ concentration, which in turn supports that GhRbohD is regulated by Ca^2+^.

In plant immunity, rbohD plants over-accumulate salicylic acid, jasmonic acid and antimicrobial compounds upon pathogen attack, such as immune marker gene expression [34,35]. To further elucidate the effect of *GhRbohD* on Verticillium wilt resistance in cotton plants, qRT-PCR was employed to monitor the expression of disease-resistance genes. In *GhRbohD*-silenced plants, as key genes of the lignin metabolism pathway, *GhCHI*, *GhPAL5* and *GhC4H1* were significantly downregulated by varying degrees (Figure 5). Generally, the higher the lignin content of the plant, the stronger the resistance to pathogen challenge, especially vascular diseases [11,36]. Besides, *GhJaz1*, an important gene of the jasmonic acid pathway, showed significantly downregulated expression and reached its lowest value at 48 hpi, accounting for just one-sixth of that of the control. Collectively, these results indicate that ROS production mediated by *GhRbohD* could induce cotton to acquire local and systemic resistance by the lignin metabolism and jasmonic acid signaling pathways.

The identification of pathogens by plants triggers several early defense responses, including the production of ROS. RBOHs are major sources of ROS during plant pathogen interactions [37,38]. *AtRbohD* and *AtRbohF* were initially regarded as key components of plant defense mediated *Arabidopsis* disease resistance [39]. Further, *AtRbohD*-dependent H_2_O_2_ signaling was a critical modulator in the defense response against Vd-toxins, regulated by the PTPs-MPKs-WRKY pathway in *Arabidopsis* against *V. dahliae* [40]. In the current study, the results of the disease resistance test showed that *GhRbohD*-silenced cotton plants showed more serious symptoms than these of the control plants, and the DI value of *GhRbohD*-silenced plants was significantly increased by 53.16% compared to the control plants (Figure 3), accompanied by a significant reduction in H_2_O_2_ content. However, when *RbohD* overexpressed in *Arabidopsis* improving the resistance to *V. dahliae*, this reduced the infection of fungus in plant roots (Figure 4). In conclusion, these results strongly support that *RbohD* positively regulates the resistance of plants against *V. dahliae*. Furthermore, Y2H assays was applied to screen the target proteins interacting with GhRbohD in cotton, to find that GhPBL9 and GhRPL12C may interact with GhRbohD. Notably, AtPBL9 is a serine threonine protein kinase with serine/threonine protein kinase activity, and the N-terminal of *GhRbohD* has four obviously phosphorylated sites; the specific phosphorylation activation process between GhRbohD and GhPBL9, and the mechanism of interaction proteins remain to be further explored.

## 4. Materials and Methods

### 4.1. Plant Material and Growth Condition

The cotton cultivar used in this experiment is *Gossypium hirsutum* L. (Zhongzhimian 2), which is resistant to *V. dahliae*, and cultivar Jimian 11 is susceptible to this pathogen. The introgression lines BC_6_F_3:6_ were generated from the advanced backcross and repeated using Zhongzhimian 2 as the recipient parent and Jimian 11 as the donor parent. Two materials NIL-R and NIL-S selected from the BC_6_F_3:6_ population were resistant and susceptible to Verticillium wilt, respectively [24]. Cotton plants for virus-induced gene silencing (VIGS) analysis were cultured in the bottomless paper pot for *V. dahliae* infection, grown in the greenhouse at 28 °C with a 16 h/8 h light/dark photoperiod.

Tobacco (*Nicotiana benthamiana*) seedlings were grown in a greenhouse at 25 °C under a 16 h/8 h light/dark photoperiod for *GhRbohD* gene subcellular localization. *Arabidopsis thaliana* were grown in a 23–25 °C greenhouse with a 16 h/8 h light cycle.

### 4.2. Fungal Strain and Growth Condition

The strong pathogenic defoliating *V. dahliae* strain Vd080 was cultured on potato dextrose agar (PDA) medium at 25 °C for 5 days. Then, this fungus was cultured in liquid Czapek-Dox media at 25 °C for 5 days, and the conidia was collected. The final concentration of the spore suspension was adjusted to 1 × 10^7^ CFU/mL with sterile water [41].

### 4.3. Gene Cloning and Bioinformatic Analyses

Total RNA was extracted from cotton roots using the RNAprep Pure Plant Plus Kit (Tiangen, Beijing, China) according to the manufacturer’s instructions. *TransScript^®^* All-in-One first-strand cDNA Synthesis SuperMix for reverse transcription kit was used to synthesize cDNA. The amplified product was cloned into the p*EASY*^®^-Blunt Cloning Kit vector, and confirmed by sequencing. The cloning primers (*GhRbohD*-full-F/R) and other primers mentioned below are listed in Appendix A.

The amino acid sequence alignment and phylogenetic relationship analysis were performed on ClustalX2.1 and MEGAX, respectively [42,43]. A total of Rbohs gene sequences were downloaded from TAIR and the COTTON database. Multiple sequence alignments of all identified Rbohs from cotton and *Arabidopsis* were performed in ClustalX2.1. The phylogenetic tree of deduced amino acid sequences was constructed by applying the neighbor-joining method. To analyze the exon-intron distributions of the Rbohs gene, the gene structure display server (GSDS) was utilized. Next, conserved motifs were predicted using the MEME9 tool [44]. The positional information of a given cotton Rbohs was obtained from parsed general feature format (GFF) files, and downloaded from the Cotton Gene website. The Rbohs in *Gossypium* species genomes were all mapped onto the chromosomes used the TBtools software [45]. The promoter sequences of Rbohs family genes were extracted and upload to PlantCARE database (http://bioinformatics.psb.ugent.be/webtools/plantcare/html/ (accessed on 8 August 2021)) for prediction of cis-acting elements. The cis-acting elements distribution upon promoters of OPR genes were displayed by GSDS v2.0 [46]. To detect expression pattern of *GhRbohD* gene, we analysed the transcriptome data of Rbohs family in *G. hirsutum* from https://cottonfgd.org/ (accessed on 12 August 2021).

### 4.4. Virus-Induced Gene Silencing (VIGS)

The specific fragments of the *GhRbohD* and positive control gene *PDS* were amplified by PCR, digested with *BamH* I and *Kpn* I and then cloned into the tobacco rattle virus (TRV) vector pYL156 to generate pYLRbohD and pYLPDS vectors. The vectors pYL156, pYLRbohD, pYLPDS and auxiliary vector pYL192 were transformed into *Agrobacterium tumefaciens* strain GV3101 and cultured in LB medium with 50 µg/mL kanamycin, 30 µg/mL gentamicin sulphate and 40 µg/mL rifampicin at 28 °C for 2–3 days. Then the *Agrobacterium* cells were resuspended in MMA solution (10 mM N-morpholino ethane-sulfonic acid, 10 mM MgCl_2_, and 200 mM acetoyringone) and adjusted to 1.2 value of OD_600_. The *Agrobacterium* cells containing the aforementioned vectors were equally mixed with those containing pYL192 and were incubated at room temperature for 3 h in darkness [47]. Finally, a needleless syringe was used to inject the mixed bacterial solution into the back of 7-days-old cotton cotyledon. The experiment was performed three times, and each sample comprised of more than 50 cotyledons.

After the albino phenotype appeared in the positive control plants, the successfully silenced plants were inoculated with the *V. dahliae* strain Vd080 at conidia suspension (1 × 10^7^ CFU/mL) by the root dipping method for 10 min and then replanted into soil.

### 4.5. qRT-PCR Analysis

The cotton leaf RNA extraction and reversed transcription into cDNA method referred to the above. Quantitative real-time PCR was performed using the LightCycler 480 System. A 20 μL reaction mixture containing diluted cDNA and TB Green Premix Ex Taq™ II (Tli RNaseH Plus) was used for qRT-PCR following the procedure: 94 °C for 2 min, followed by 45 cycles of 94 °C for 15 s, 55 °C for 15 s and 72 °C for 20 s. qRT-PCR was performed to quantify the transcript levels of several disease resistance-related genes, with *UBQ* gene as internal control [48]. The expression assays were repeated three times and each assay was performed with three independent technical repeats. The relative expression levels of genes were calculated using the 2^−ΔΔCt^ method [49].

### 4.6. Plant Disease Resistance Assess

A scale of 0–4 was used to classify plants including *GhRbohD* gene silenced cotton and transgenic *Arabidopsis* according to the percentage of plant tissue affected by chlorosis, leaf necrosis or defoliation (0: no symptoms, 1: ≤33%, 2: >33% and ≤66%, 3: >66% and ≤99%, 4: ≥99% leaves with chlorosis wilt symptoms). The disease index (DI) was calculated as previously described, DI = [(0n_0_ + 1n_1_ + 2n_2_ + 3n_3_ + 4n_4_)/4n] × 100, where n_0_-n_4_ were the numbers of plants with each of the corresponding disease ratings, and n was the total number of plants assessed [50]. Futhermore, *V. dahliae* recovery assay was applicated to determine the effects of *V. dahliae* infection on cotton plants.

### 4.7. Observation of ROS, Lignin Synthesis and Callose Accumulation

To verify ROS by observing the accumulation of H_2_O_2_, the leaves from the control and *GhRbohD*-silenced cotton plants were collected at 24 h, 48 h after inoculation with *V. dahliae* Vd080. After incubation in a large centrifuge tube with an appropriate amount of 3, 3-Diaminobenzidime (DAB, 1 mg/mL, pH 7.5) at room temperature in darkness for 8 h, leaves were decolorized in 95% ethanol for 2 min, followed by decolorizatiion in absolute ethanol until the green of the leaves was completely removed. Then accumulation of H_2_O_2_ of leaves was observed under a microscope in 70% glycerol [51,52].

Lignification of the control and *GhRbohD*-silenced plants was examined with phloroglucinol staining. Root sections of cotton seedlings were incubated in 10% phloroglucinol solution for 2 min. The samples were then incubated in concentrated H_2_SO_4_ for a moment, and the staining signals were observed using a microscope [5].

Callose staining used the true leaves of the control and *GhRbohD*-silenced cotton. Leaves were fixed in fixative solution (ethanol: acetic acid = 3:1) for 2 h to remove chlorophyll, then soaked in 70% and 50% ethanol for 2 h, respectively, and put in water overnight. After rinsing the leaves with water, they were treated in 10% NaOH for 1 h to make them transparent. The callose content was observed under a fluorescent microscope with UV excitation light after being cultured in darkness in 0.01% aniline blue for 3 h [5]. Each experiment was repeated three times.

### 4.8. Visualization of H_2_O_2,_ NO and Ca^2+^

A Quantitative Assay Kit (Nanjing Jiancheng, Beijing, China) was used for the determination of H_2_O_2_ and nitric oxide (NO) in the control and *GhRbohD*-silenced cotton plants, referring to the manual for detailed operation steps.

The calcium content of the control and *GhRbohD*-silenced plants’ leaves were determined with a Calcium Colorimetric Assay Kit (Nanjing Jiancheng, Beijing, China). The samples’ absorbance was measured at 575 nm, and the concentrations of calcium were calculated based on the formula: C = S_a_/S_v_ (S_a_: amount of calcium in unknown sample (µg) from standard curve; S_v_: sample volume (µL) added into the wells; C: concentration of calcium in sample). Three independent biological and technical repeats were performed.

### 4.9. Subcellular Localization

The full-length coding sequence of *GhRbohD* (without stop codon) was inserted into the overexpression vector p*CAMBIA3300* with GFP tag to construct the transient expression vector *GhRbohD*-GFP. Then it was transformed into *Agrobacterium tumefaciens* GV3101. The mixture of GV3101 bacteria solution of *GhRbohD*-GFP and buffer solution were injected into the leaves of 3-weeks-old tobacco plants at the stage of 4–5 leaves with a syringe. Then the tobacco was cultured in darkness for 24 h and then transferred to normal conditions for 24 h. Finally, the fluorescence in the leaves could be observed with a laser confocal microscope [8].

### 4.10. Arabidopsis thaliana Transformation

The full length CDs of *GhRbohD* were cloned by using cotton cDNA as template, and the *GhRbohD* gene fragment was connected to the p*CAMBIA2300* vector with a 35S strong promoter by digesting the sequence with cision enzyme. Next, the recombinant plasmid was transformed into *Agrobacterium tumefaciens* GV3101 with the freeze thaw method. The overexpression vector was transformed into *A. thaliana* Col-0 via the floral dip method. The transformants (T_0_, T_1_ and T_2_ seeds) were screened for survival on a half concentration of MS medium containing 50 mg/L kanamycin. T_3_ transgenic plants were identified with qRT-PCR analysis.

The seeds of T_3_ transgenic lines were inoculated on medium (1/2 MS), and after these seedlings were grown for about two weeks, the roots were directly inoculated with 5 µL Vd080 conidia suspension (1 × 10^7^ CFU/mL) [53]. Then *Arabidopsis thaliana* were cultured in the soil for disease resistance identification.

### 4.11. Yeast Two-Hybrid Assays (Y2H)

The *GhRbohD* cDNA was amplified using the primer pair *GhRbohD-BD-F/R*. PCR-amplified fragments were cloned in-frame with GAL4BD in the vector p*GBKT7*. BD-GhRbohD was introduced into the yeast strain Y2HGold. The cDNA library was constructed from upland cotton roots inoculated with *V. dahliae*, aimed to Y2H screen with BD Matched maker Library Construction & Screening Kits (Clontech). Then they were co-transferred into yeast receptive cells and cultured on the DDO (SD/-Leu-Trp) culture medium for one to two weeks. If there were any clones, these were transferred to the QDO (SD/-His-Leu-Trp-Ade) medium for verification.

## 5. Conclusions

From our previous study, we found early differentially expressed protein GhRbohD by phospho-proteomic analysis, which responded to plant defenses against pathogens. In this study, the phylogenetic and structural analysis of Rbohs by bioinformatics analysis indicated that the Rbohs family was conserved in cotton and *A. thaliana* plants. In detail, there were 12 members of RbohD out of 57 Rbohs in cotton. VIGS test demonstrated that ROS production, NO, H_2_O_2_ and Ca^2+^ contents of *GhRbohD*-silenced cotton plants were significantly reduced, as the same as lignin synthesis and callose accumulation. In addition, expression profile analysis showed that *GhCHI*, *GhPAL5*, *GhC4H1* and *GhJaz1* were significantly downregulated, involving the lignin metabolism and jasmonic acid signaling pathways. Further investigation showed that the overexpressed *Arabidopsis* exhibited stronger resistance. In conclusion, these results provide evidence of the role of *GhRbohD* mediated ROS production, confirming the positive regulation of *GhRbohD* in plants against *V. dahliae*.

## Figures and Tables

**Figure 1 ijms-22-13041-f001:**
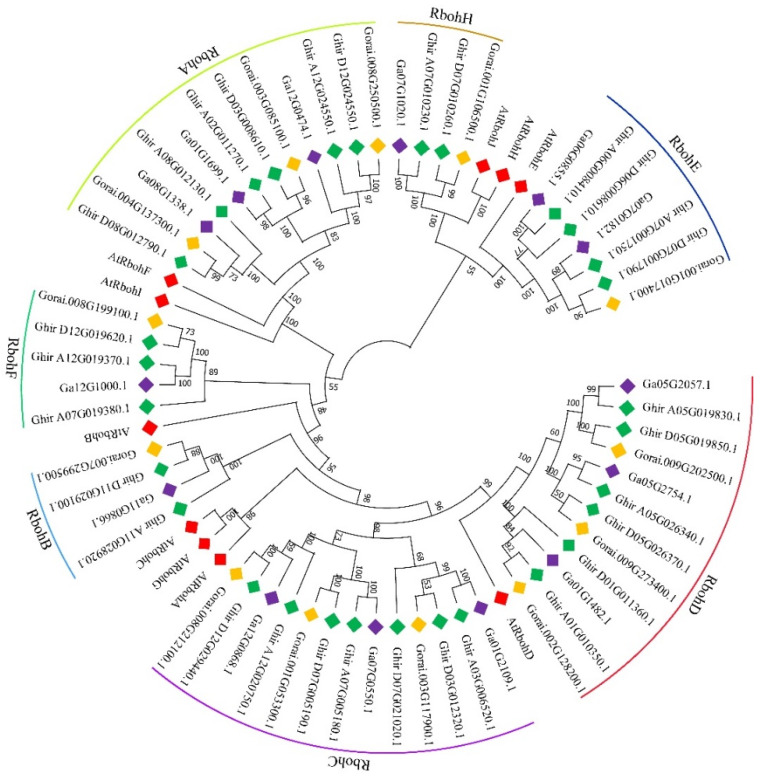
Phylogenetic analysis of Rbohs in *Gossypium* species. A phylogenetic tree of Rboh proteins from *G. arboreum*, *G. raimondii*, *G. hirsutum* and *A. thaliana*. The full-length amino acid sequence of Rbohs protein family was compared by ClustalX in MEGA7.0 using the neighbor-joining (NJ) method. Different colored solid squares indicate different genes of the genus Cotton or *Arabidopsis*.

**Figure 2 ijms-22-13041-f002:**
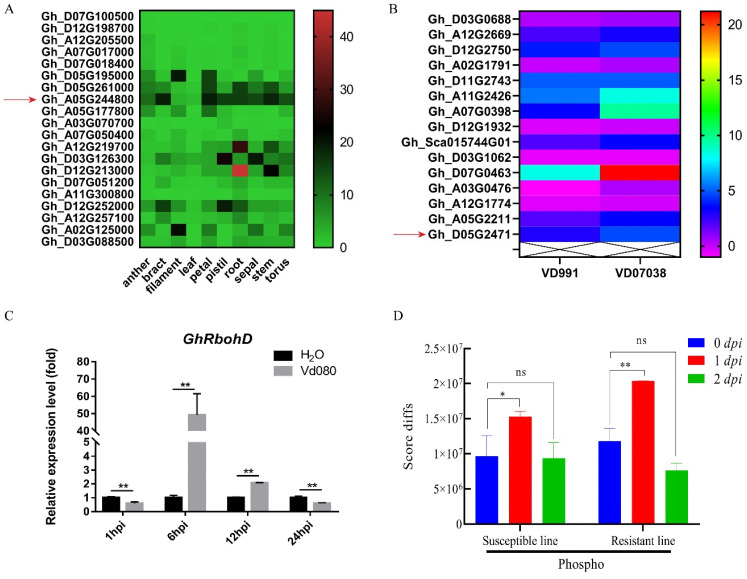
The expression of *GhRbohD* in cotton under different conditions. (**A**) Transcriptional level analysis of GhRbohD family genes in various tissues of Upland cotton YZ-1. (**B**) The transcription levels of GhRboh family genes in *G. hirsutum* under strong pathogenicity (VD991) and moderate pathogenicity (VD07038) stress. (**C**) The expression level of *GhRbohD* in Zhongzhimian 2 under pathogen stress. (**D**) Analysis of the levels of phosphorylation modification of proteins after translation at different times RIL-R and RIL-S varieties under pathogenic stress [24]. Asterisks indicate statistically significant differences, as determined by Student’s *t*-test (* *p* < 0.05; ** *p* < 0.01). ns represents no significant difference.

**Figure 3 ijms-22-13041-f003:**
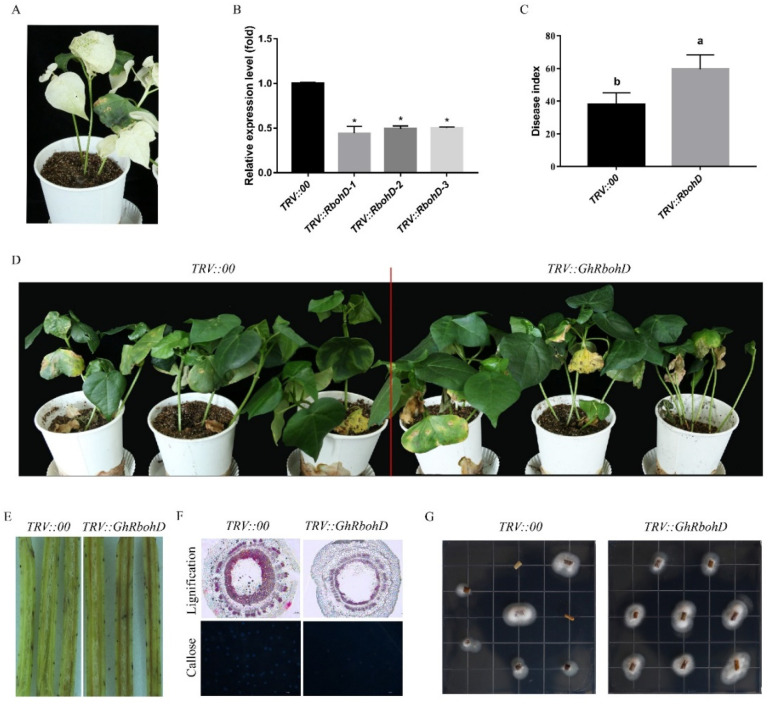
The resistance of cotton to *Verticillium dahliae* decreased after *GhRbohD* silencing. (**A**) TRV::PDS as the positive control for silencing efficiency. Albino phenotype appeared at 12 days after infection. (**B**) The expression level of *GhRbohD* in silenced plants and control plants. (**C**) Assessment of DI for *TRV::00* plants and *TRV::RbohD* plants at 20 dpi. (**D**) The incidence of disease in control and silencing plants. (**E**) Vascular browning in cotton stem after infection. (**F**) The xylem stain in stem and callose detection in leaf. (**G**) *Verticillium dahliae* recovery assay. Asterisks indicate statistically significant differences, as determined by Student’s *t*-test (* *p* < 0.05). Different letters a and b represent significant differences.

**Figure 4 ijms-22-13041-f004:**
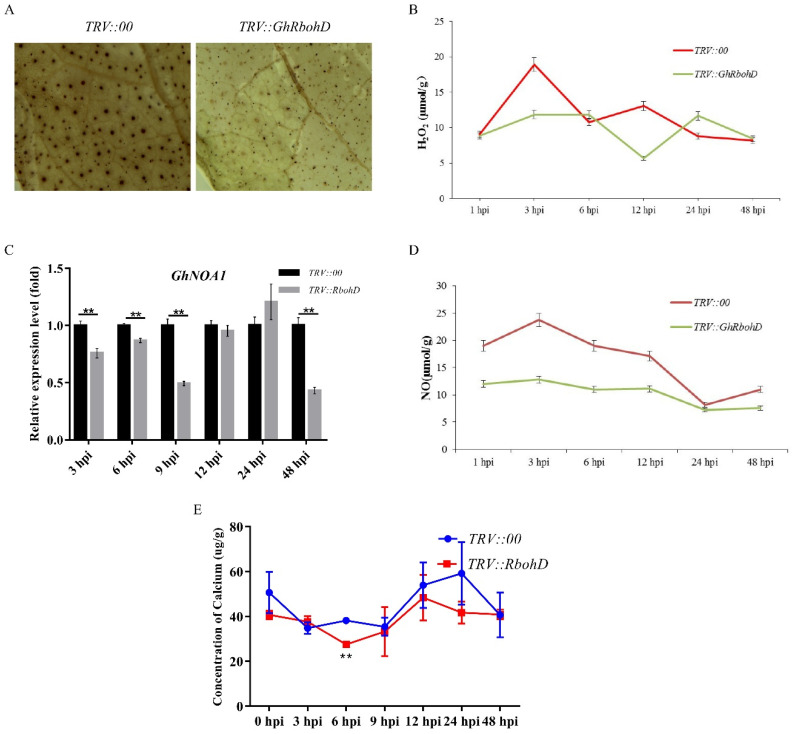
*GhRbohD* loss-of-function plants has reduced ROS production level. (**A**) Detection of ROS in cotton leaves. (**B**) Determination of H_2_O_2_ content. (**C**) The expression of *GhNOA1*. (**D**) Determination of NO content. (**E**) Determination of calcium content. Asterisks indicate statistically significant differences, as determined by Student’s *t*-test (** *p* < 0.01).

**Figure 5 ijms-22-13041-f005:**
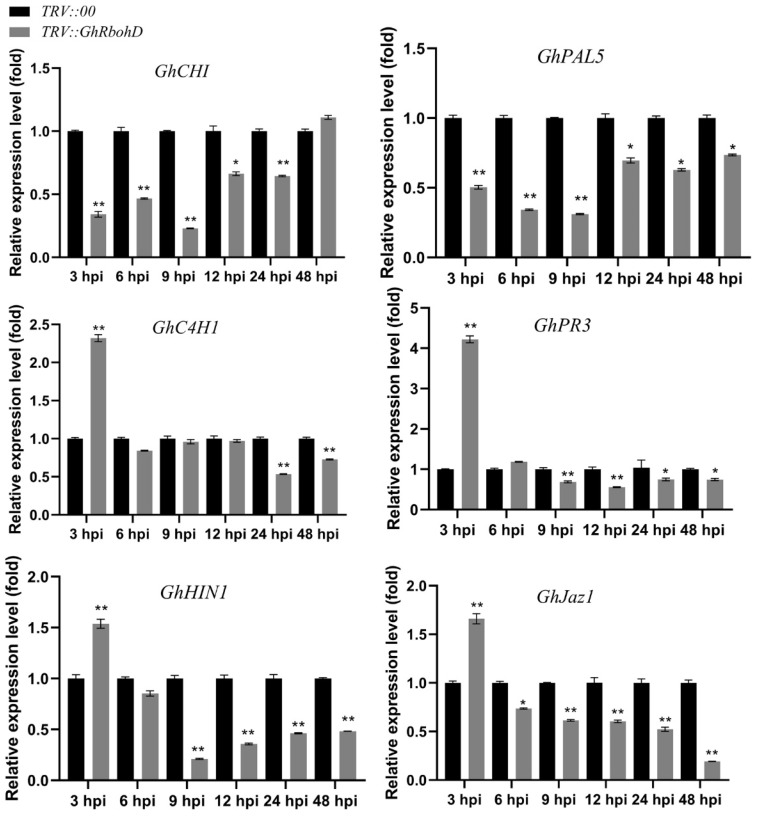
Expression of disease-resistant genes in *GhRbohD* silencing and control plants. Error bars represent the standard deviation of three biological replicates. Asterisks indicate statistically significant differences, as determined by Student’s *t*-test (* *p* < 0.05; ** *p* < 0.001).

**Figure 6 ijms-22-13041-f006:**
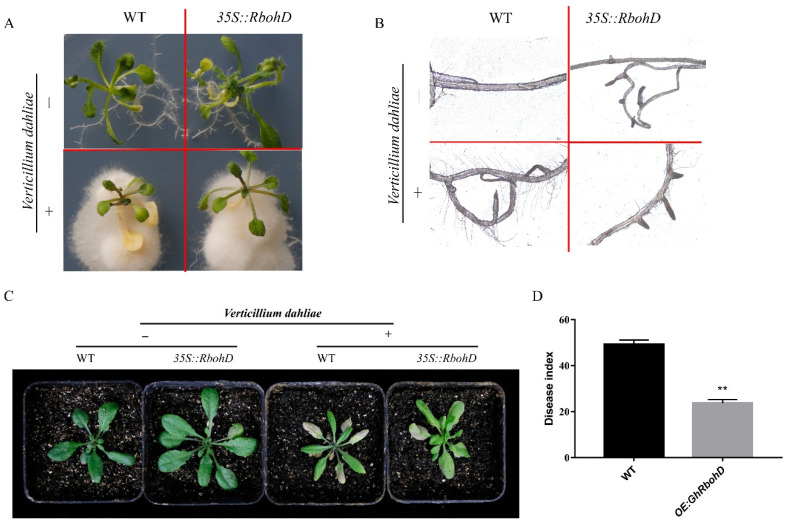
Overexpressing *GhRbohD Arabidopsis thaliana* enhanced the resistance to *V. dahliae*. (**A**) Disease resistance phenotypes of WT and OE-GhRbohD *Arabidopsis* inoculated with *V. dahliae* in MS culture medium. (**B**) The amount of mycelium attachment around the roots. (**C**) Disease resistance phenotypes of WT and OE-GhRbohD *Arabidopsis* inoculated with *V. dahliae*. (**D**) The DI of WT and OE-GhRbohD *Arabidopsis*. Asterisks indicate statistically significant differences, as determined by Student’s *t*-test (** *p* < 0.01).

**Figure 7 ijms-22-13041-f007:**
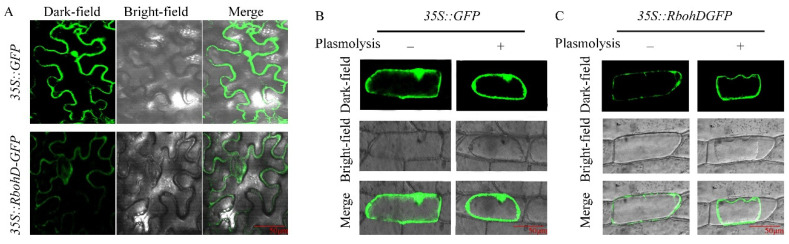
Subcellular localization of GhRbohD. (**A**) Subcellular localization of 35S::GhRbohD-GFP fusion proteins were transiently expressed in *N. benthamiana* leaves. (**B**,**C**) 35S::GhRbohD-GFP fusion proteins were transiently expressed in onion epidermal cells. The signal was visualized with confocal microscopy. Scale bar = 50 μm.

**Figure 8 ijms-22-13041-f008:**
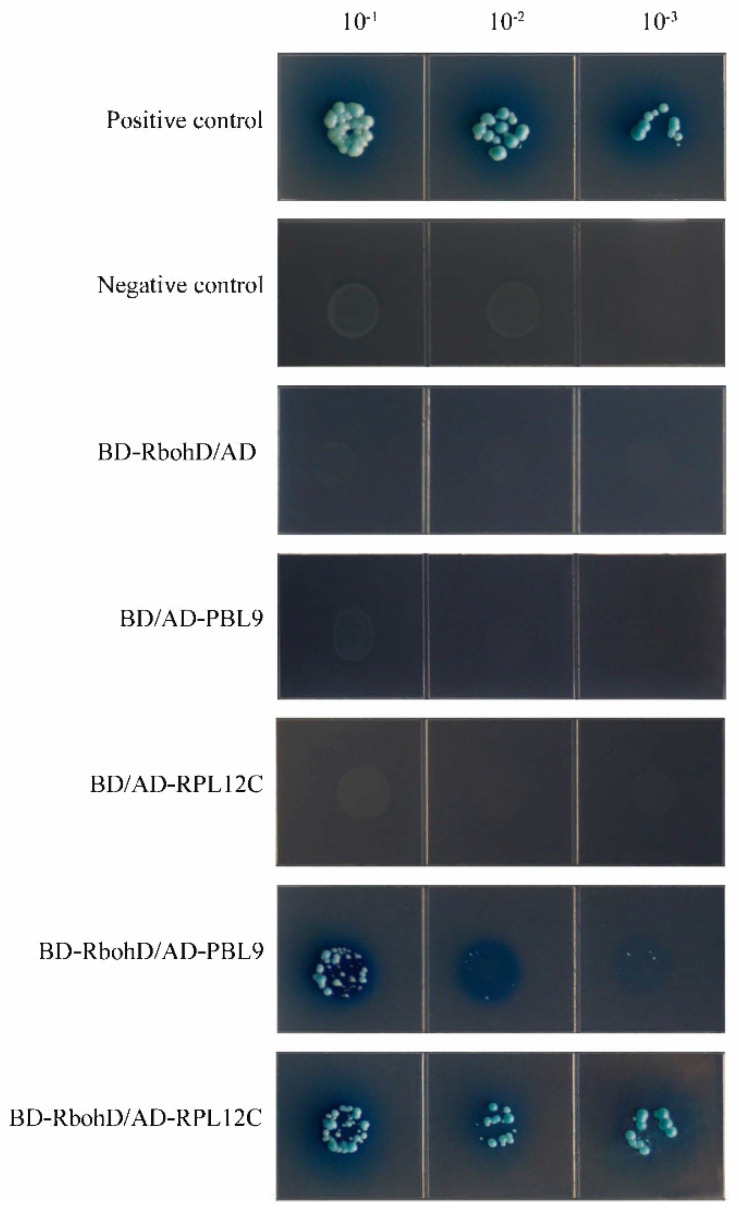
Yeast two-hybrid assays of the interactions of GhRbohD with GhPBL9 and GhRPL12C. Transformants were grown on SD/-Leu/-Trp/-Ade/-His (+X-α-gal) media. pGBKT7-53/pGADT7-RecT was used as the positive control. pGBKT7-Lam/pGADT7-RecT were used as negative controls.

## Data Availability

The data used to support the findings of this study are available from the corresponding author upon request.

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
