# Peer review of "The Respiratory Burst Oxidase Homolog Protein D (GhRbohD) Positively Regulates the Cotton Resistance to Verticillium dahliae"

_ijms, 2021, doi:10.3390/ijms222313041_

Round 1
Reviewer 1 Report
The manuscript entitled “The respiratory burst oxidase homolog protein D (GhRbohD) 2 positively regulates the cotton resistance to Verticillium dahlia” showed evidences about the role of GhRbohD protein and mediated by them ROS level in resistance of Gossypium species. The complexity and comprehensiveness of the research performed here is impressive. The way the research were conducted, their description as well as the discussion conducted convinces the reader of the correctness of the conclusions. The publication have potential to give high impact on improving Gossypium resistance to pathogens infection. The experiments were done I recomment the manuscript for publication and have only few minor comments:
In material and methods should be mentioned the susceptible line of cotton described in section 2.2 and figure 2D
Proof of the resistance of the cotton cultivar Zhongzhimian 2 used in the experiments should be given in section 4.1 or supplementary material
ROS is related to plant homeostasis in control conditions, I am wondering how looks like the TRV::RbohD plants at 20 dpi in control condition, is it normal growth and development for such plants? (as in Figure 3 D)
The conclusion about conservativeness of Rbohs family among plant kingdom should be more careful as it based on phylogenetic analyzes carried out only on a selected plant family.
How to explain such drastic increasing of GhRbohD gene expression after 6 hpi, is it related to some specific plant reaction? It should be mention in section 3.
How to comment that the silencing method is low efficient as can be deduced from Figure 3 B? Can it have an influence on all conclusions from experiments?
The sentence in line 270: “Numerous evidences showed that the identification of plant earlier resistance genes was one of the most effective ways to prevent Verticillium wilt” should be rewritten.
Line 335: Nicotiana benthamiana
Author Response
Comment 1: In material and methods should be mentioned the susceptible line of cotton described in section 2.2 and figure 2D.
Our response: Thank you for your careful observation. We have supplemented the corresponding contents.
Comment 2: Proof of the resistance of the cotton cultivar Zhongzhimian 2 used in the experiments should be given in section 4.1 or supplementary material.
Our response: Yes, we introduced the corresponding reference.
Comment 3: ROS is related to plant homeostasis in control conditions, I am wondering how looks like the TRV::RbohD plants at 20 dpi in control condition, is it normal growth and development for such plants? (as in Figure 3 D)
Our response: The growth and development of the TRV::RbohD plants were inhibited and more sensitive to pathogens.
Comment 4: The conclusion about conservativeness of Rbohs family among plant kingdom should be more careful as it based on phylogenetic analyzes carried out only on a selected plant family.
Our response: Yes, we have revised it.
Comment 5: How to explain such drastic increasing of GhRbohD gene expression after 6 hpi, is it related to some specific plant reaction? It should be mention in section 3.
Our response: The drastic increasing of GhRbohD gene expression after 6 hpi indicated that this gene may play an important role in the response of cotton to the stress of V. dahliae.
.
Comment 6: How to comment that the silencing method is low efficient as can be deduced from Figure 3 B? Can it have an influence on all conclusions from experiments?
Our response: In this paper, the gene silencing efficiency was about 0.5, which was relatively low. This may be related to gene structure and genetic transformation system. Compared with the control, plants with low silencing efficiency can also explain these results.
Comment 7: The sentence in line 270: “Numerous evidences showed that the identification of plant earlier resistance genes was one of the most effective ways to prevent Verticillium wilt” should be rewritten.
Our response: Yes, we have rewritten it “Numerous evidences showed that the functional identification of plant earlier resistance genes was a promising way to prevent Verticillium wilt”
Comment 8: Nicotiana benthamiana
Our response: Thanks, we have corrected it.
Reviewer 2 Report
Some indirect evidence is presented that suggests GhRbohd "...positively regulates the cotton resistance to Verticillium dahliae" as indicated in the title, but the overexpression experiments are understandably conducted on model crop Arabidopsis thaliana. Conclusions are careful to implicate resistance in "plants" instead of specifically cotton, but the title should probably reflect the same care.
Verify Figure 2 is from or extracted from the previous study, and if so, please cite where previously published in the figure title. The susceptible and resistant cultivars used are not mentioned in this manuscript Materials nor supplemental materials provided.
Provide some justification and discussion that the albino phenotype used in gene silencing procedure did not affect plant response to V. dahliae infection.
In the first sentence of Discussion section, "...the main reason is the lack of disease-resistant germplasm resources [6,7]" a review of those interesting articles does not appear to implicate lack of resistant germplasm resources but rather a lack of understanding of the mechanisms to effectively utilize molecular and genetic engineering techniques to further develop resistant cultivars using advanced techniques. Research underlying further advancement in knowledge (including previous study to this report) makes use of 'resistant' and 'susceptible' germplasm so it is assumed the germplasm resources are available, just not effectively used since the resistance mechanisms are most likely quite complex and not simple gene for gene interactions. Resistance to diseases like Verticillium wilt is more complicated than single gene system (i.e. gene-for-gene models); where, the phenotypic response is usually discrete (immunity or a high level of resistance). Most cotton diseases are better explained by a quantitative model with a continuous degree of phenotypic response to infection, where a gene confers incremental increases in resistance.
Nice paper.
Author Response
Comment 1: Some indirect evidence is presented that suggests GhRbohd "...positively regulates the cotton resistance to Verticillium dahliae" as indicated in the title, but the overexpression experiments are understandably conducted on model crop Arabidopsis thaliana. Conclusions are careful to implicate resistance in "plants" instead of specifically cotton, but the title should probably reflect the same care.
Our response: Thanks for your constructive advice, because we pay more attention to the role of GhRbohD in cotton resistance to Verticillium wilt, it may be more appropriate to use cotton in the title.
Comment 2:
Verify Figure 2 is from or extracted from the previous study, and if so, please cite where previously published in the figure title. The susceptible and resistant cultivars used are not mentioned in this manuscript Materials nor supplemental materials provided.
Our response: Yes, we have supplemented the corresponding contents.
Comment 3: Provide some justification and discussion that the albino phenotype used in gene silencing procedure did not affect plant response to V. dahliae infection.
Our response: Albino phenotype plant was only used to observe whether VIGS system was normal, but not to inoculate V. dahliae.
Comment 4: In the first sentence of Discussion section, "...the main reason is the lack of disease-resistant germplasm resources [6,7]" a review of those interesting articles does not appear to implicate lack of resistant germplasm resources but rather a lack of understanding of the mechanisms to effectively utilize molecular and genetic engineering techniques to further develop resistant cultivars using advanced techniques. Research underlying further advancement in knowledge (including previous study to this report) makes use of 'resistant' and 'susceptible' germplasm so it is assumed the germplasm resources are available, just not effectively used since the resistance mechanisms are most likely quite complex and not simple gene for gene interactions. Resistance to diseases like Verticillium wilt is more complicated than single gene system (i.e. gene-for-gene models); where, the phenotypic response is usually discrete (immunity or a high level of resistance). Most cotton diseases are better explained by a quantitative model with a continuous degree of phenotypic response to infection, where a gene confers incremental increases in resistance.
Our response: We admire your professional opinion and revise the relevant contents.